# The fluorination effect of fluoroamphiphiles in cytosolic protein delivery

Zhenjing Zhang[1], Wanwan Shen[1], Jing Ling[1], Yang Yan[1], Jingjing Hu[1] & Yiyun Cheng[1]

Direct delivery of proteins into cells avoids many drawbacks of gene delivery, and thus has emerging applications in biotherapy. However, it remains a challenging task owing to limited charges and relatively large size of proteins. Here, we report an efficient protein delivery system via the co-assembly of fluoroamphiphiles and proteins into nanoparticles. Fluorous substituents on the amphiphiles play essential roles in the formation of uniform nanoparticles, avoiding protein denaturation, efficient endocytosis, and maintaining low cytotoxicity. Structure-activity relationship studies reveal that longer fluorous chain length and higher fluorination degree contribute to more efficient protein delivery, but excess fluorophilicity on the polymer leads to the pre-assembly of fluoroamphiphiles into stable vesicles, and thus failed protein encapsulation and cytosolic delivery. This study highlights the advantage of fluoroamphiphiles over other existing strategies for intracellular protein delivery.

---

[1] Shanghai Key Laboratory of Regulatory Biology, School of Life Sciences, East China Normal University, Shanghai 200241, China. Correspondence and requests for materials should be addressed to Y.C. (email: yycheng@mail.ustc.edu.cn)

Numerous human diseases arise from mutations or other abnormalities on proteins owing to their essential functions in enzyme catalysis, signal transduction, gene regulation, maintaining the delicate balance between cell survival, and programmed cell death, etc[1,2]. During the past decades, protein biotherapeutics including peptide hormones, growth factors, cytokines, and monoclonal antibodies have been discovered or engineered to treat these diseases[3]. Protein therapeutics offer several advantages over small molecule drugs, such as higher specificity, limited adverse effects, and faster clinical development[4]. However, proteins are generally membrane-impermeable due to their relatively large size, hydrophilicity and limited positive charges, which make them difficult to reach the intracellular targets[5]. Therefore, most current protein drugs are developed based on extracellular targets. To expand the family of protein therapeutics, it is of great importance to develop efficient strategies for cytosolic protein delivery[6,7].

Cytosolic protein delivery is challenging since the vehicle should efficiently bind the protein, protect it against degradation, initiate efficient internalization, trigger endosomal disruption, and release the proteins into cytosol[8]. In retrospect, the most studied approach has been fusing proteins with protein transduction domains (PTD)[9–12]. However, PTD-based strategies displayed a certain number of limitations that most of them require chemical modification of target proteins[13]. The covalent conjugation of PTD to cargo proteins might be involved with reduced bioactivity and safety concerns. Besides, many other delivery systems have been developed based on liposomes[14,15], peptides[16], polymers[17,18], and inorganic nanoparticles[19,20]. These approaches still possess some limitations such as the need of protein modification, complicated synthesis, and limited transduction efficacy.

Recently, fluoroamphiphiles such as fluorinated dendrimers and polyethylenimines (PEIs) were reported to have promising features like no other non-viral carriers[21–25]. These materials encountered multiple hurdles during gene delivery. In addition, the fluoroamphiphiles possess excellent self-assembly property[22,26–31]. Combining these features together, it is rational to develop fluoroamphiphiles for cytosolic protein delivery. Proteins could be fabricated into nanoparticles via the co-assembly of fluoroamphiphiles and proteins. The fluorocarbons could improve the affinity of polymers to cell membranes and facilitate the endocytosis[32–34]. In addition, the fluorous ligands are generally lipophobic and bioinert, and this property is beneficial for the avoidance of protein denaturation and the retention of protein bioactivity. Finally, the replacement of hydrocarbons on traditional amphiphiles with fluoroalkyls is responsive for less cytotoxicity and hemolytic activity[35].

As a proof-of-concept, we synthesize a small library of fluoroamphiphiles by grafting fluoroalkyls to branched PEI for cytosolic protein delivery. Traditional amphiphiles such as alkane- and cycloalkane-grafted PEIs are also included to reveal the effect of fluorination and highlight the advantages of fluoroamphiphiles over other existing amphiphilic materials. Model proteins such as bovine serum albumin (BSA), β-galactosidase (β-Gal), saporin, and peptide (GRKKRRQRRREKIKRPRSSNAETL) with different molecular sizes and charge properties (BSA and β-Gal are negatively charged, saporin and the peptide are positively charged) are employed to test the efficacy of developed fluoroamphiphiles. We demonstrate that the discovered fluoroamphiphiles efficiently deliver unmodified proteins into cells without inducing cytotoxicity.

## Results

**Screening of efficient fluoroamphiphiles.** Branched PEI was grafted with fluoroalkanes (F1-F4), alkanes (A1-A4), and cycloalkanes (C1-C4) via amine-epoxide or amine-isocyanate reactions (Fig. 1)[36,37]. Each ligand was coupled to PEI at four conjugation degrees. Take F3 for example, the average numbers of conjugated F3 per PEI were 28, 54, 76, and 102, respectively, and the materials were termed F3-1, F3-2, F3-3, and F3-4, respectively. Unmodified PEI and a commercial reagent Pulsin$^{TM}$ were used as negative and positive controls, respectively. A total number of 50 materials were used as the screening pool to discover efficient materials (Supplementary Table 1).

We first investigated efficacies of the 50 candidates in the library using a fluorescein isothiocyanate labeled BSA (BSA-FITC, 0.3 μM). The highest fluorescence intensity for each material was shown in Supplementary Fig. 1 and Table 1. The transfected cells at optimal condition for each material were also treated with trypan blue to quench the BSA-FITC physically adsorbed on cell membrane[38]. The fluorescence intensity of cells after trypan blue quenching was shown in Fig. 1c.

As shown in Fig. 1c, 11 materials (7 fluoroalkane-grafted and 4 alkane-grafted materials) showed superior efficacies to the positive control Pulsin$^{TM}$. The top two performing fluoroamphiphiles F4-1 and F3-3 are much more efficient than the others in the library. When F4-1 and F3-3 mediated BSA-FITC delivery was carried out at 4 °C, or added with sodium azide, fluorescence from the cells decreased significantly (Supplementary Fig. 2), suggesting endocytosis as the primary mechanism of internalization. The endocytosis of F4-1 and F3-3 complexes is mediated by both macropinocytosis- and caveolae-dependent pathways (Supplementary Fig. 3). The cells treated with F4-1/F3-3 complexes exhibited strong and evenly distributed fluorescence in the cytosol after 1 h incubation, and the internalized proteins were not co-localized with acidic organelles (Supplementary Fig. 4). These results proved that F4-1 and F3-3 are capable of rapidly transporting proteins into cells and releasing cargos in the cytosol.

**Fluorination effect of fluoroamphiphiles.** Since the efficacies of F4-1 and F3-3 are much superior to the non-fluorinated analogs. We further investigated the effect of fluorination on cytosolic protein delivery. It was reported that the contribution of a $CF_2$ group to hydrophobicity was about 1.5 times that of a $CH_2$ group[39], and thus the hydrophobicity of F3 and F4 could be roughly equivalent to that of A3 and A4, respectively. In this case, A4-1 and A3-3 in the library can be used as non-fluorinated controls for F4-1 and F3-3, respectively. As revealed by the confocal images in Fig. 2a, BSA-FITC delivered by F4-1 or F3-3 were mainly dispersed in cytosol, while those by A4-1 and A3-3 were observed in green dots and seemed to be absorbed on cell membrane. The fluorescence quenching experiment in Fig. 2b further confirmed this hypothesis, nearly 90% fluorescence intensity from F4-1/F3-3 complexes was retained after the addition of trypan blue, a cell membrane-impermeable fluorescence quencher[38], while the values were less than 40% for non-fluorinated controls. Even when the transfection time for non-fluorinated polymers was increased from 4 to 24 h, the observed fluorescence was scarcely increased, and only green dots were observed on cell surface (Supplementary Figs. 5 and 6). Both non-fluorinated complexes are internalized via macropinocytosis- and caveolae-dependent pathways (Supplementary Fig. 7). Fluorocarbons are both hydrophobic and lipophobic, and the mixing of fluorocarbons and hydrocarbons is highly non-ideal[40]. Therefore, the fluoroamphiphiles and phospholipids have limited miscibility,

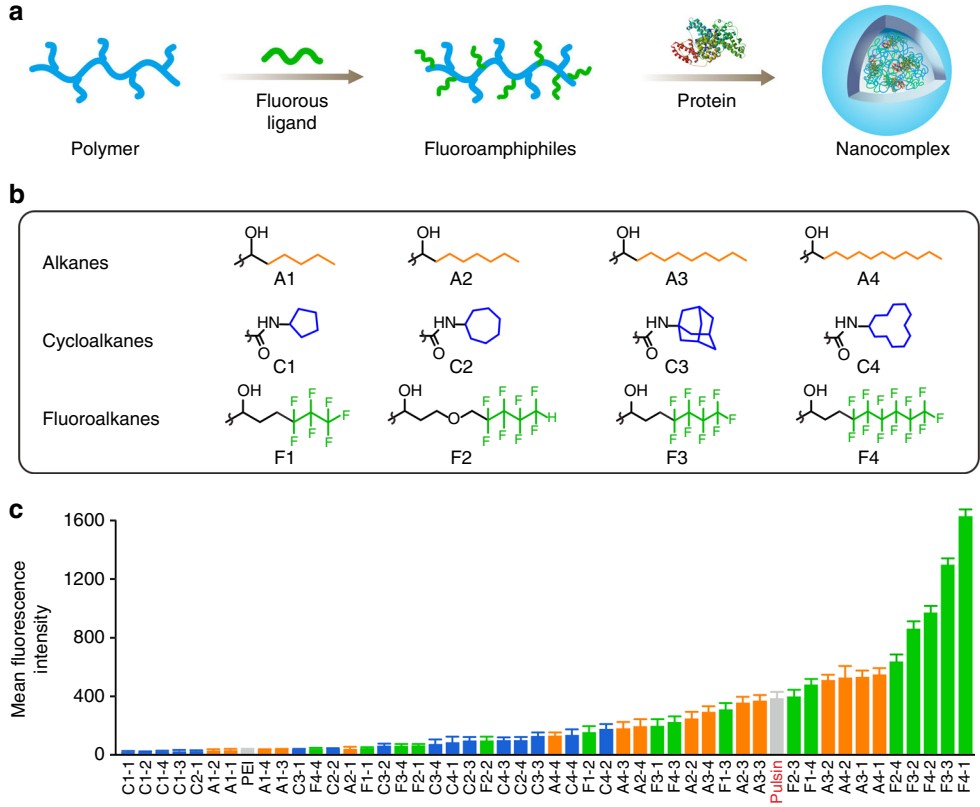

**Fig. 1** Fluoroamphiphiles for cytosolic protein delivery. **a** Co-assembly of fluoroamphiphiles and proteins. **b** Structures of hydrophobic substituents coupled to PEI. A1-A4 alkanes, C1-C4 cycloalkanes, F1-F4 fluoroalkanes. **c** Mean fluorescence intensity of cells transfected with nanocomplexes after trypan blue treatment. Data are presented as the mean ± s.e.m. ($n = 3$)

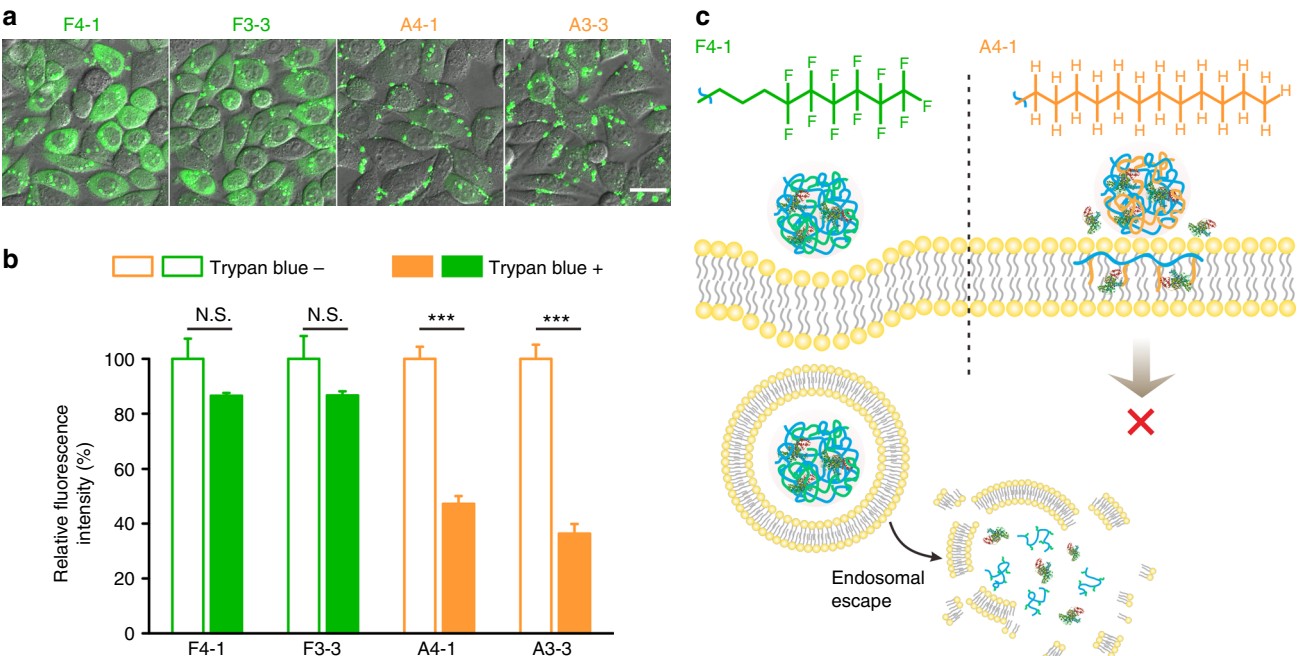

**Fig. 2** Fluoroamphiphiles vs. non-fluorinated controls. **a** Confocal images of HeLa cells treated with polymer/BSA-FITC complexes for 4 h. The polymers were used at optimal doses listed in Supplementary Table 1. The scale bar is 25 μm. **b** Relative fluorescence intensity of cells in **a** before and after trypan blue quenching. The fluorescence intensity of cells treated without trypan blue was defined as 100%. Data are presented as mean ± s.e.m. ($n = 3$). N.S.$p > 0.05$ and ***$p < 0.001$ analyzed by Student's $t$-test, one tailed. **c** Proposed mechanisms of F4-1 and A4-1 in cytosolic protein delivery

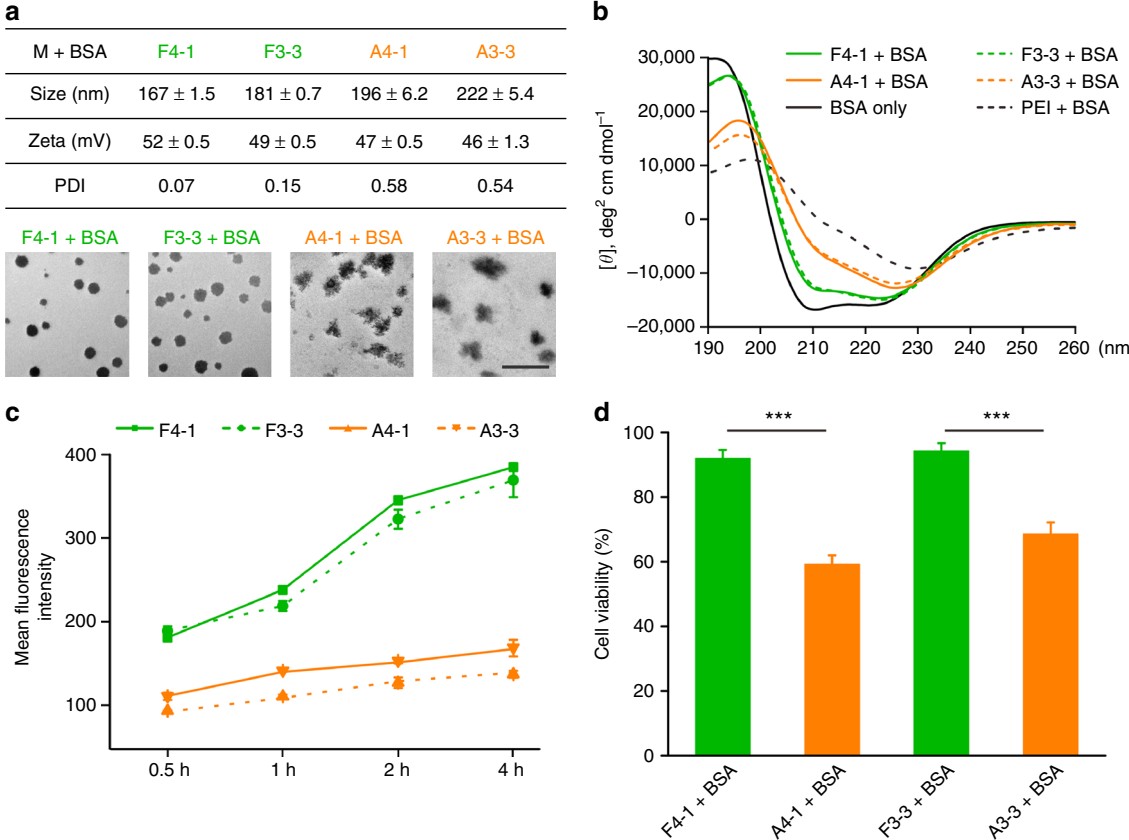

**Fig. 3** Mechanisms of fluoroamphiphiles. **a** Size, zeta-potential, PDI, and TEM images of nanocomplexes. Data are presented as mean ± s.d. ($n = 3$). A representative result from three independent experiments. The scale bar is 500 nm. **b** Circular dichroism spectra of nanocomplexes. BSA concentration in **b** is 3 μM, and the polymer/protein molar ratio in **a**, **b** is 1:1. **c** Cellular uptake of FITC-labeled polymers in the absence of proteins. The polymer concentration is 5 μg/mL. Data are presented as mean ± s.e.m. ($n = 3$). **d** The viability of HeLa cells treated with different nanocomplexes. The BSA concentration is 0.3 μM. Data are presented as mean ± s.e.m. ($n = 5$). ***$p < 0.001$ analyzed by Student's $t$-test, one tailed

which ensures efficient cell internalization of fluoroamphiphiles[41,42]. In comparison, the non-fluorinated controls are consisted of hydrocarbon chains, which are easily fused with the membranes after cell attachment[43]. The fusion of phospholipids and hydrogenated amphiphiles will lead to early release of proteins and failed cytosolic delivery (Fig. 2c). It is worth noting that all the cycloalkane-based amphiphiles showed poor efficacies, and the materials were even less efficient than alkane-based ones. BSA-FITC delivered by cycloalkane amphiphiles, such as C4-2 also showed significant membrane absorption (Supplementary Fig. 8). These results suggested the beneficial effect of fluorination in cytosolic protein delivery.

We further compared the behaviors of fluoroamphiphiles and non-fluorinated controls in the aspects of nanocomplex formation, protein denaturation, cellular uptake and cytotoxicity. As shown in Fig. 3a, fluoroamphiphiles and BSA formed uniform nanoparticles (~200 nm) with a low particle dispersion index (PDI) around 0.1, while the non-fluorinated A4-1 and A3-3 tend to form anomalous aggregations (PDI > 0.5). This result can be explained by the strong surface activity of fluoroamphiphiles compared with non-fluorinated ones. It was reported that the incremental change in the free energy of adsorption for the transfer of a $CF_2$ group from water to the air/water surface is almost twice that of a $CH_2$ group ($-5.1$ vs. $-2.6$ kJ/mol)[44]. The strong surface activity of fluoroamphiphiles dramatically increases the tendency to assemble in water[26,27,40,45]. In comparison, hydrogenated amphiphiles possess relatively lower surface activity, and the lipophilic chains also have high affinity

with proteins via hydrophobic interactions, which drives the formation of large aggregates. The non-specific hydrophobic interactions between alkanes and proteins may denature the bound proteins. Circular dichroism results in Fig. 3b confirmed this hypothesis. BSA complexed with A4-1 and A3-3 showed significant changes in protein secondary structures, while those bound by fluoroamphiphiles relatively approached native BSA. The fluoroalkyl chains are relatively bioinert and usually used for antifouling purposes[46–49], therefore the F4-1 and F3-3 nanocomplexes showed limited protein denaturation. The partially denatured BSA in the A4-1 or A3-3 complexes also explained the long-term absorptions of BSA-FITC on cell membranes in Fig. 2a and Supplementary Fig. 5[50].

Next, we tested the membrane permeability of fluoroamphiphiles in the absence of proteins. As shown in Fig. 3c, fluoroamphiphiles labeled with FITC showed much higher internalization than non-fluorinated controls, which is in accordance with results observed on fluorinated dendrimers[21,51,52]. Fluoroalkyl chains have a strong tendency to adsorb on cell membranes due to exceedingly low fluoroalkyl-water interactions[53], and favorable cell adhesion is beneficial for efficient cellular uptake[43]. In addition, the limited miscibility between fluoroalkyl and phospholipids can minimize the fusion of amphiphiles with cell membranes during endocytosis, which is a dominant feature for hydrogenated amphiphiles[54]. Fusion of hydrogenated amphiphiles not only leads to long-term retention of materials on cell membrane, but may also cause membrane disruption and cytotoxicity[43]. Therefore, the fluoroamphiphiles

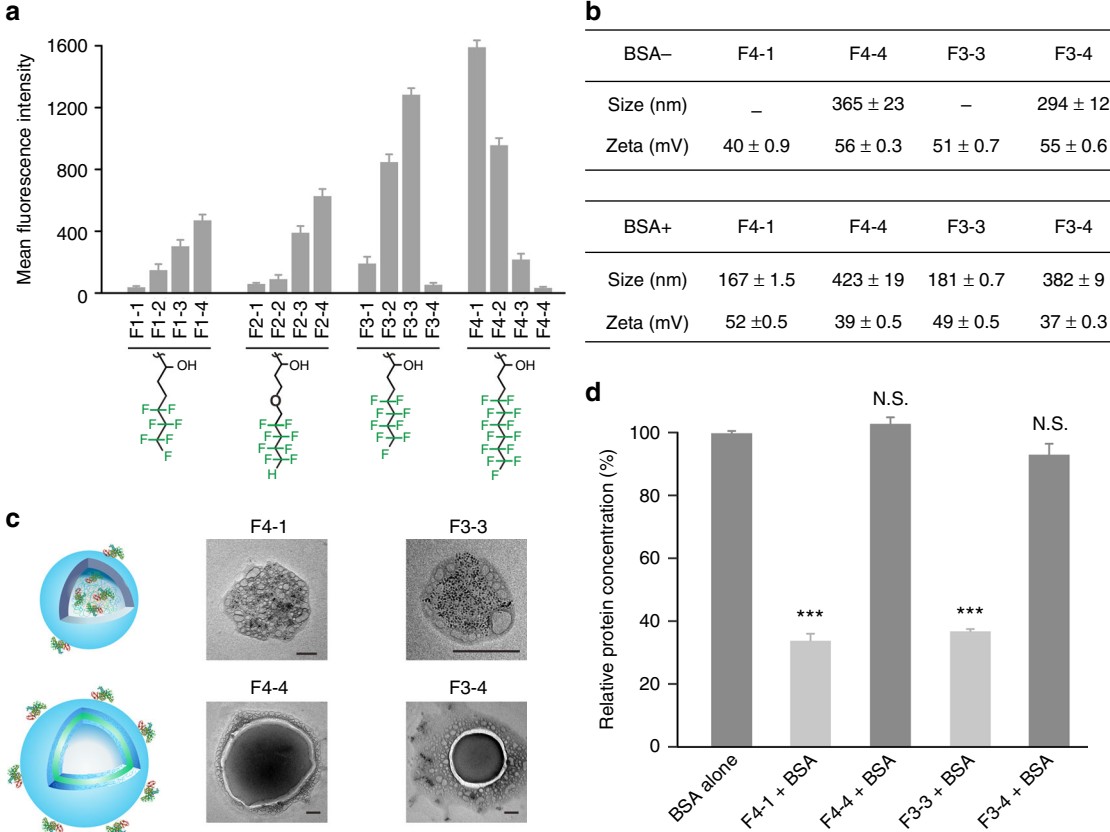

**Fig. 4** Structure-activity relationships of fluoroamphiphiles. **a** Mean fluorescence intensity of HeLa cells transfected with nanocomplexes for 4 h at optimal conditions. The cells were treated with trypan blue before flow cytometry measurement. Data are presented as mean ± s.e.m. ($n = 3$). **b** Size and zeta-potential of fluoroamphiphiles and their complexes. Data are presented as mean ± s.d. ($n = 3$). A representative result from three independent experiments. **c** TEM images of fluoroamphiphiles complexed with Pt-labeled BSA. The polymer to protein molar ratio in **b** and **c** is 1:1. The scale bar is 100 nm. **d** BCA assay for the nanocomplexes. Free BSA was tested as a control. Data are presented as mean ± s.e.m. ($n = 5$), $^{N.S.}p > 0.05$ and $^{***}p < 0.001$ analyzed by Student's $t$-test, one tailed

and their complexes showed much lower toxicity than the hydrogenated controls (Fig. 3d and Supplementary Fig. 9).

**Structure-activity relationships of fluoroamphiphiles**. We further investigated the structure-activity relationships of fluoroamphiphiles in cytosolic protein delivery. As shown in Fig. 4a, the fluoroamphiphiles with longer chains or higher grafting degrees generally exhibited higher efficacies. However, this rule is invalid for F3-4, F4-3, and F4-4. These fluoroamphiphiles possess relatively long fluoroalkyl chains and high fluorination degrees, but showed extremely low delivery efficacy. It is known that longer fluoroalkyl chains and higher fluorination degrees lead to higher gene delivery[55]. Considering differences between protein and gene delivery, we hypothesized that the low efficacies of F3-4, F4-3, and F4-4 are attributed to poor protein encapsulation. In this case, we systemically investigated the self-assembly behaviors of fluoroamphiphiles. As shown in Fig. 4b and Supplementary Fig. 10, all the fluoroamphiphiles except F3-4, F4-3, and F4-4 failed to assemble into nanoparticles in the absence of BSA at concentrations up to 15 μM, while F3-4, F4-3, and F4-4 could form nanoparticles or vesicles (PDI < 0.3) even at 0.3 μM (Supplementary Fig. 11). As previously reported, a decrease of hydrophilic to hydrophobic segment ratio in amphiphiles lead to the change of assembled nanostructures from spherical to cylindrical micelles and finally to vesicles[56]. The assembly of fluoroamphiphiles usually forms internal Teflon-like hydrophobic and lipophobic films that increase the stability of assembles and reduces its permeability to hydrophilic molecules[40,53].

This process may hinder the encapsulation of protein within the assembled nanostructures, and thus proteins only bind on the surface of nanoassembles via electrostatic interactions. As a result, the complexation of these fluoroamphiphiles with BSA, an anionic protein at pH 7.4, leads to the decrease of zeta-potential of formed nanoparticles. We further labeled BSA with platinum (Pt) nanoparticles[57]. The TEM results in Fig. 4c confirmed that the encapsulation of Pt-labeled BSA was hindered in F3-4 and F4-4 nanoassembles. On the other hand, F3-3 and F4-1 co-assembled with BSA to form nanocomplexes (PDI < 0.3). The binding of anionic region of BSA to cationic PEI facilitate the assembly process. The element mapping results in Supplementary Fig. 12 showed that Pt nanoparticles were homogeneously distributed in the F4-1/F3-3 nanocomplexes. The failed protein encapsulation by F3-4 and F4-4 is further confirmed by a BCA assay. As shown in Fig. 4d, nearly 100% BSA complexed with F3-4 and F4-4 could be detected by the BCA assay, while 70% proteins within the F4-1 and F3-3 complexes could not be measured, which is an indication of protein encapsulation. These results together confirmed that excess fluorophilicity on the polymer leads to pre-assembly of fluoroamphiphiles and failed protein encapsulation.

**Robustness of fluoroamphiphiles**. We further tested the efficacies of F4-1 and F3-3 on other cell lines. As shown in Fig. 5a, b, both materials successfully delivered BSA-FITC into NIH3T3 and HEK293 cells. In addition, fluoroamphiphile-mediated BSA-FITC delivery is much more efficient than TAT-conjugated proteins (Supplementary Fig. 13). The materials are also efficient in the

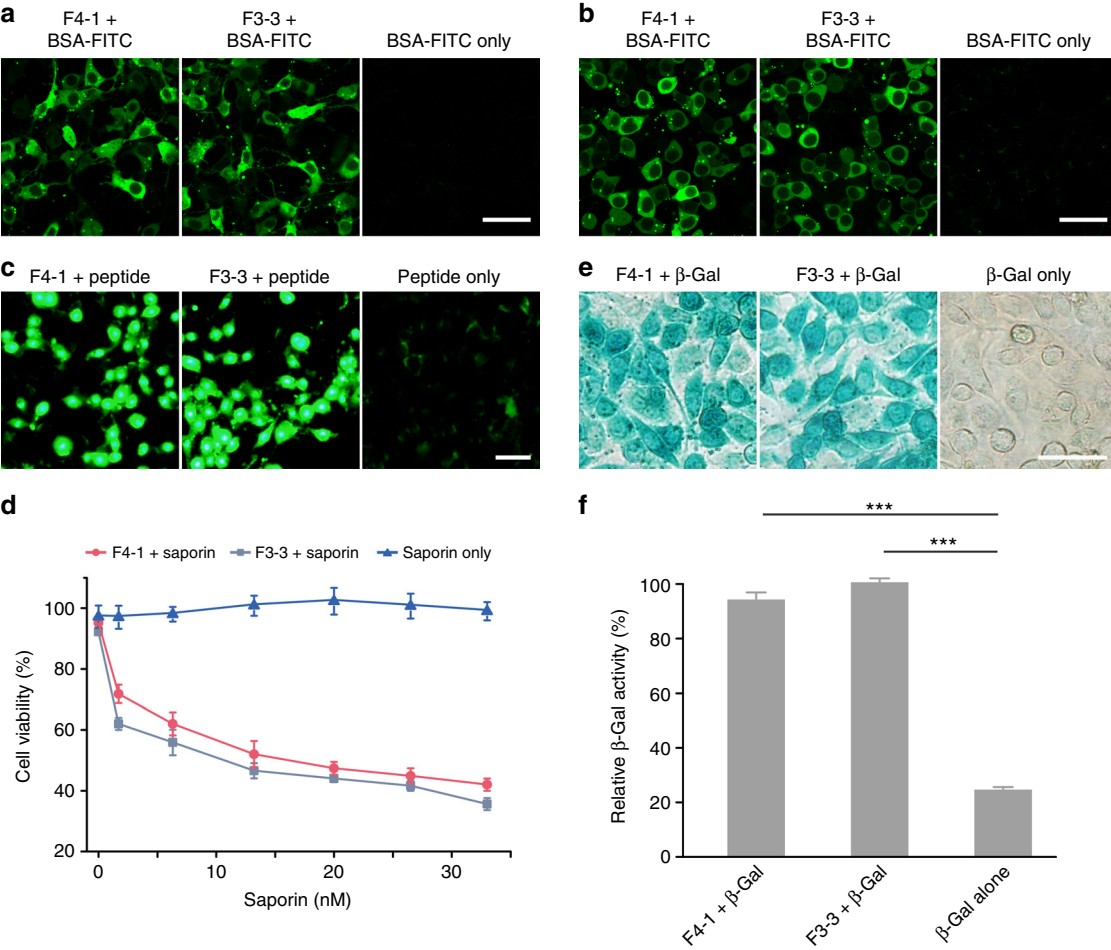

**Fig. 5** Fluoroamphiphiles in protein delivery. Confocal images of NIH3T3 (**a**) and HEK293 (**b**) cells treated with the nanocomplexes for 4 h. **c** Fluorescence images of HeLa cells transfected with the complexes for 4 h. A volume of 4 μg BSA-FITC or 1 μg peptide-FITC was complexed with 2 μg F4-1 and 2.5 μg F3-3, respectively. A representative result from three independent experiments. **d** Concentration-dependent toxicity of saporin and fluoroamphiphile/saporin complexes on HeLa cells. Data are presented as mean ± s.e.m. ($n = 5$). X-gal staining (**e**) and relative β-Gal activity (**f**) of HeLa cells treated with the complexes for 4 h. 5 μg β-Gal, 2 μg F4-1, or 2.5 μg F3-3 were used. A representative result from three independent experiments for **e**, data are presented as mean ± s.e.m. ($n = 6$) for **f**. ***$p < 0.001$ analyzed by Student's $t$-test, one tailed. The scale bar in the figure is 50 μm

delivery of β-Gal, saporin, and a peptide (GRKKRRQRRRE-KIKRPRSSNAETL-FITC). As shown in Fig. 5c–f, F4-1 and F3-3 showed impressive efficacies in the delivery of these biomolecules. More importantly, the bioactivity of proteins such as β-Gal delivered by the fluoroamphiphiles could be maintained (Fig. 5f and Supplementary Fig. 14). These results confirmed that F4-1 and F3-3 can be used as vehicles for the delivery of various proteins without the need of chemical modification.

## Discussion

In the library, the fluoroalkanes F1, F2, F3, and F4 have 7, 8, 9, and 13 fluorine atoms, respectively. The striking difference between F1/F2- and F3-based fluoroamphiphiles in protein delivery can be explained by an additive fluorination effect. A single or two fluorine differences in the structure may have significant influence on its physicochemical properties and transduction efficacy. Therefore, the length of fluoroalkyl chain and the fluorination degree on a specific fluoroamphiphile need to be optimized before use. Structure-activity relationship studies reveal that a balance in fluorophilicity is beneficial for efficient protein delivery. Pre-assembly of the fluoroamphiphiles before protein encapsulation should be avoided. The lead fluoroamphiphiles F4-1 and F3-3 discovered in the library successfully

delivered various proteins with distinct isoelectric points and molecular weights into the cytosol. The fluoroamphiphiles showed robustness of nanoparticle formation with BSA, β-gal, and saporin at different weight ratios (Supplementary Table 2). Though saporin is a positively charged protein at physiological conditions[58], the binding of anionic region on saporin to the cationic fluoroamphiphiles may help the formation of nanocomplexes. The nanocomplexes formed at different protein to amphiphile weight ratios were tested on cells to confirm the effectiveness. The fluorescence from HeLa cells increased in proportion to protein concentration in the range of 2–20 μg/mL, and the internalization is saturated at higher protein concentrations (Supplementary Fig. 15), suggesting dose-dependent protein delivery. Though fluorinated substances were listed as persistent and bioaccumulative materials[59], the concentrations of fluoroamphiphiles in cytosolic protein delivery are minute, and further design of biodegradable fluoroamphiphiles may resolve the bioaccumulation issues for in vivo protein delivery.

In conclusion, we found the fluorination effect of polymers in cytosolic protein delivery. The fluoroamphiphiles show advantages in several aspects including improved protein encapsulation, avoiding protein denaturation, facilitated cellular uptake, and limited material toxicity in comparison with non-fluorinated

materials. The fluoroamphiphiles allow the delivery of proteins into cells without the need of protein modification.

## Methods

**Intracellular protein delivery**. BSA-FITC (4 μg) was mixed with the amphiphiles at different material to protein weight ratios. The yielding nanocomplexes were diluted with 50 μL serum-free media and incubated at room temperature for 30 min. Then the complexes were replenished with 150 μL serum-free media and added to the cells. After 4 h, the media were removed and the cells were washed with PBS and analyzed by flow cytometry (BD FACSCalibur, San Jose) and laser scanning confocal microscope (LSCM, Leica SP5, Germany). In a separate study, trypan blue (0.2 mg/mL) was added to the transfected cells before flow cytometry measurement. Pulsin$^{TM}$ was used according to the manufacture's protocol (4 μL reagent for each well). For other proteins, 5 μg β-Gal or 1 μg peptide were complexed with 2 μg F4-1 or 2.5 μg F3-3, respectively. The transfection procedure was the same as described above, and the cells were observed by fluorescence microscope (Olympus, Japan).

Materials and other methods are available in Supplementary Methods.

**Data availability**. The data supporting the findings in this study are available within the article and its supplementary information files. All data are available from the authors upon reasonable request.

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

## Acknowledgements

We thank financial supports from National Natural Science Foundation of China (21725402 and 21474030) and Shanghai Municipal Science and Technology Commission (17XD1401600).

## Author contributions

Z.Z. conducted most experiments on material screening, protein delivery, analyzed the data, and wrote the manuscript. W. S. synthesized and characterized the polymers. J.L., Y. Y., and J.H. performed part of complex characterization and protein delivery. Y.C. designed and supervised the study and wrote the manuscript.

## Additional information

**Competing interests:** The authors declare no competing interests.

