## [Peer Review File(PDF 3142 kb) · Nature Communications]

Reviewers' comments:

Reviewer #1 (Remarks to the Author):

The efficient cytosolic delivery of proteins is still a major challenge in biomedical research and drug development. Here, the authors demonstrate the non-covalent complexation of proteins with fluoro-substituted amphiphiles as a solution to this problem. Remarkably, fluoro-substituted amphiphiles are more effective in achieving import than the hydrogen-containing counterparts. The results are presented using a series of complementary techniques. I appreciate that next to delivery, also the structure of the complexes is investigated by size measurements and electron microscopy, including element mapping with which stoichiometric information can be obtained.

Overall, the concept is interesting and may certainly be worth of publication once the points listed below have been addressed which will require some major revisions.

A main problem of the presentation of results is the inconsistency in relating to transmembrane transport and endocytosis followed by endosomal release. In particular, this is apparent at ll. 51-52 "improve the affinity of polymers to cell membranes and the penetration of polymers across the lipid bilayers, facilitating endocytosis and endosomal escape". So is uptake by endocytosis the presumed mode of action and then cytosolic delivery occurs by enhanced cytosolic release? I miss a careful time lapse analysis of import into cells to delineate the mechanism of action. At least the experiments using endocytosis inhibitors suggest import by endocytosis.

The authors claim that their delivery system has excellent activity. However, in the way the data is presented it is impossible to validate this claim. What are the concentrations of proteins present in the incubation solutions? It would be very helpful, if a comparison of the carriers to an established carrier system such as a Tat-conjugate would be included.

I would appreciate if the applicability to other cell lines was presented.

Even though the authors apply their system to two proteins and two peptides, complex formation is investigated only for BSA. Information on nanoparticle formation and the robustness of nanoparticle formation to different ratios should be presented. How much effort is needed to adapt the delivery approach to a particular experimental system?

The cellular delivery of beta-Gal followed by X-gal staining is not a proof for the full preservation of activity. The staining could as well result from a minor fraction of non-denatured protein. Would it be feasible to titrate beta-Gal with the amphiphile and determine the maintenance of activity in an in vitro system?

For the proteins, also a concentration series for delivery is required. It could be that at low concentrations, the complexes disassemble before cellular entry. On the other hand, is it guaranteed that the complexes dissociate once inside the cell?

The element mapping analysis would provide the means to understand to which degree complexation is a function of the molar ratio of the two components.

Minor points:

Abstract, l. 11: instead of Fluorous ligands it should read fluoruous substituents.

Introduction: l. 37: I do not know of evidence that CPP-mediated delivery requires the denaturation of the cargo protein. Moreover, CPPs have been presented that work by non-covalent complexation (Biochim Biophys Acta. 2010 Dec;1798(12):2274-85

). Thus covalent modification is not required in all cases.

Reviewer #2 (Remarks to the Author):

Review on the manuscript entitled:

"The fluorous effect of fluoroamphiphiles in cytosolic protein delivery"
by Zhang et al.

This paper reports on the potential of fluorinated amphiphiles in the cytosolic delivery of proteins. The fluorinated amphiphiles are co-assembled to form nanoparticles in aqueous solutions. This work is based on the fact that the combined hydrophobic and lipophobic characters of fluorinated chains improve the affinity of polymers for cell membranes and their penetration across lipid bilayers, thus facilitating the endocytosis for protein delivery.

A series of fluoroamphiphiles was obtained by grafting various fluoroalkyl chains to a cationic poly(ethylamine) (PEI) polymer, and their capacity to deliver proteins and peptides (bovine serum albumin, β -galactosidase, saporin and the peptide GRKKRRQRRREKIKRPRSSNAETL) was investigated. Their ability to deliver these molecules was compared to that of traditional amphiphiles such as alkane and cycloalkane-grafted PEI. The results show that the fluoroamphiphiles have definite advantages over the non-fluorinated amphiphiles, such as improved protein encapsulation, lower protein denaturation, facilitated cellular uptake and limited toxicity. A structure-activity relationship investigation also revealed that a balance in the fluorophilic character of the fluoroamphiphiles is beneficial for efficient protein delivery. The paper is clearly organized and written, and adequately referenced. The methods are appropriate, and the results carefully interpreted.

The only objection that could be raised is that fluorinated amphiphiles degrade eventually into carboxylic acids that are persistent to the environment and toxic (see Krafft and Riess, *Curr. Opin. Colloid Interface Sci.* 2015, 20, 192). The concentrations used in this particular application are minute however, but the authors may consider adding a comment in order to lift a potential objection.

To my opinion this work provides a clear breakthrough in the field and should interest the broad readership of *Nature Communications*. I recommend publication with minor modifications.

Typos:

Line 57: "fluoroamphiphiles"

Line 72: "cycloalkanes"

Figure 1 caption: "cycloalkane"

Reviewer #3 (Remarks to the Author):

In the manuscript "The fluorous effect of fluoroamphiphiles in cytosolic protein delivery", Cheng and coworkers report the preparation of fluoroamphiphile/protein nanocomplexes for effective protein delivery into cells. Their characterization of these assemblies and the structure-activity relationship studies are thorough and make for a complete paper. The data presented in Figure 2 clearly show their strategy is successful. The use of fluorous amphiphiles for protein delivery is novel, and is appropriate for publication in *Nature Communications* once the following comments are addressed:

Library Synthesis and Initial Comparisons

- It should be explicitly stated in the text that branched PEI was used to make the various polymer amphiphiles. Figure 1a should also be modified to show the branched structure of the polymer

being modified with the fluororous ligands. It should also more realistically schematize the nanocomplex.

- Number of conjugated fluororous ligands is given as a number average. It would be useful for weight % fluorine to be provided in Figure 5 or Table S1, as this is standard for characterizing fluororous compounds and makes for easier comparison between the different materials in the F series.
- If known, it would be useful to describe/show the general structure of commercial PULSin™ and how it compares to their polymer material.
- The use of trypan blue as a quenching agent for the extracellular fluorescence should be referenced or, if a novel procedure, experimentally validated

Cell Culture and Biomolecule Delivery

- Delivery was nicely optimized with BSA-FITC, and then translated to other enzymes to show activity after delivery. However, since cytosolic protein delivery is being claimed in the paper and the title, the authors should provide insight into how their material escapes the endosome and how the protein is released from the polymer. If the experiments in Figure 2 are meant to depict endosomal escape, that should be explicitly stated in the text. Endosomal escape could also be shown via co-localization with an endosomal stain.
- Insight into how the fluoroamphiphile cationic polymers efficiently encapsulate positively charged molecules (saporin and the peptide) should be included.
- The peptide sequence should be listed in the Figure 2 legend.

Comparison of Fluoroamphiphiles and Non-Fluorinated Controls in Protein Delivery

- In line 116, the authors write that “the fluoroamphiphiles and phospholipids in cell membranes have limited miscibility, which ensures efficient cell internalization of fluoroamphiphile-based nanocomplexes.” This should have a citation (K. Kumar work is relevant)

Nanocomplex Formation, Protein Denaturation, Cellular Uptake, and Cytotoxicity of Fluoroamphiphiles and Non-Fluorinated Controls and Structure-Activity Relationships

- The zeta-potentials of A3-3 and A4-1 nanocomplexes should be added to Figure 4a.
- The difference between materials F2 and F3 (a single hydrogen) should be commented on as the effect on protein delivery is striking.
- This discussion would greatly be enhanced with a schematic showing the proposed assembly structures. (F4-1 and F3-3 compared to F4-4 and F3-4 in Figure 5c).

Other comments

- The term “fluorous effect” which is used in the title and throughout the paper is an existing term in the literature with a specific definition referring to the fact that highly fluorinated molecules “exhibit an unusual propensity to phase segregate[e]” (Marsh, E.N.G. *Acc. Chem. Res.* 2014, 47, 2878–2886; Marsh, E.N.G. and coworkers, *Biochemistry* 2004, 43, 16277-16284) or to the “observation that highly fluorinated or perfluorinated compounds have a tendency to exclude themselves from both aqueous and organic phases” (Clark, A.W. and coworkers, *Chem. Commun.* 2017, 53, 3094; Horváth, I.T.; Curran, D.P.; Gladysz, J.A. *Handbook of Fluorous Chemistry*, Wiley-VCH Verlag GmbH & Co. KGaA, 2005, pp. 1–4).

The authors should refrain from redefining “the fluororous effect” as they do in line 101, and perhaps instead use the wording “the effect of fluorination....”

- The work in this paper would be better presented by beginning with the structures of the nanocomplexes and how those affect protein denaturation, membrane permeability, and cytotoxicity. Figure 4 should come before Figure 3 (characterization before application).

-Figure 2 is the most impressive figure, which shows how effective their scaffold is; as such, it should be the final figure of the paper.

-There are numerous typos and grammar errors, which should be corrected.

Response to the Reviewers' Comments

Reviewer #1 (Remarks to the Author):

The efficient cytosolic delivery of proteins is still a major challenge in biomedical research and drug development. Here, the authors demonstrate the non-covalent complexation of proteins with fluoro-substituted amphiphiles as a solution to this problem. Remarkably, fluoro-substituted amphiphiles are more effective in achieving import than the hydrogen-containing counterparts. The results are presented using a series of complementary techniques. I appreciate that next to delivery, also the structure of the complexes is investigated by size measurements and electron microscopy, including element mapping with which stoichiometric information can be obtained.

Overall, the concept is interesting and may certainly be worth of publication once the points listed below have been addressed which will require some major revisions.

Replies: Thanks very much for your nice comments on our manuscript! We have replied to your detailed comments one-by-one as follows.

1. A main problem of the presentation of results is the inconsistency in relating to transmembrane transport and endocytosis followed by endosomal release. In particular, this is apparent at ll. 51-52 “improve the affinity of polymers to cell membranes and the penetration of polymers across the lipid bilayers, facilitating endocytosis and endosomal escape”. So is uptake by endocytosis the presumed mode of action and then cytosolic delivery occurs by enhanced cytosolic release? I miss a careful time lapse analysis of import into cells to delineate the mechanism of action. At least the experiments using endocytosis inhibitors suggest import by endocytosis.

Replies: We agree with you that the description looks inconsistency in relating to transmembrane transport and endocytosis in the original manuscript. According to your suggestion, we conducted further experiments on the cell internalization, as well as time lapse analysis on BSA delivery. As shown in Figure R1, the internalizations of F4-1/BSA and F3-3/BSA complexes were significantly inhibited when the transfection experiments were carried out at 4 °C, or the cells were incubated with sodium azide, an ATP depleting poison and known inhibitor of endocytosis. These results implicate endocytosis is the primary mechanism of internalization for fluoroamphiphile/protein complexes. Detailed studies using particular endocytosis pathway inhibitors including chlorpromazine, cytochalasin-D and genistein showed that the latter two inhibitors significantly blocked the endocytosis (Figure R2), suggesting the internalization was mediated by macropinocytosis- and caveolae-dependent pathways.

To avoid confusion, we have removed the descriptions on “improve...the penetration of polymers across the lipid bilayers” in the manuscript. In addition, the above results and discussions on the endocytosis were added to the revised main manuscript (Supplementary Fig. 2 and Fig. 3).

Besides the endocytosis pathway, we investigated the time-dependent localization

of BSA-FITC within the cells delivered by F4-1 and F3-3, as well as non-fluorinated control materials A4-1 and A3-3. As shown in Figure R3, HeLa cells treated with fluoroamphiphile/BSA-FITC showed strong and evenly distributed fluorescence in the cytosol after 1 h incubation, and the internalized proteins (green) were not co-localized with acidic organelles (red). In comparison, cells incubated with the control complexes exhibited weak and un-internalized green dots on the cell surface. The results suggested that fluoroamphiphiles such as F4-1 and F3-3 are capable of rapidly transporting proteins into the treated cells and releasing the cargos into cytosol after endocytosis. According to your suggestion, the supplemented result on time-dependent intracellular trafficking of the complexes was added to the revised manuscript (Supplementary Fig. 4 and Fig. 6). We appreciate the valuable suggestion from the reviewer on this issue to improve our manuscript.

Figure R1. Relative fluorescence from HeLa cells treated with F4-1/BSA-FITC and F3-3/BSA-FITC complexes under various conditions. The concentration of NaN₃ is 100 mM. ***p<0.001 analyzed by Student's t-test, one tailed.

Figure R2. Internalization pathways of F4-1/BSA-FITC and F3-3/BSA-FITC complexes by HeLa cells. The concentration of chlorpromazine, cytochalasin-D and genistein are 20 μM, 10 μM and 700 μM, respectively. N.S. p > 0.05 and ***p < 0.001 analyzed by Student's t-test, one tailed.

Figure R3. Confocal images of HeLa cells treated with different polymer/BSA-FITC complexes for 0.5 h, 1 h, 2 h, 4 h, 6 h and 8 h, respectively. The acidic organelles in HeLa cells were stained with LysoTracker Red.

2. The authors claim that their delivery system has excellent activity. However, in the way the data is presented it is impossible to validate this claim. What are the concentrations of proteins present in the incubation solutions? It would be very helpful, if a comparison of the carriers to an established carrier system such as a Tat-conjugate would be included.

Replies: Thanks very much for your comments! The BSA-FITC concentration in the incubation solutions is 20 $\mu\text{g}/\text{mL}$ (4 μg in each well). We further tested efficacies of the lead materials F4-1 and F3-3 at different doses of BSA-FITC. The protein concentration ranges from 2 to 40 $\mu\text{g}/\text{mL}$. As shown in Figure R4, the fluorescence from transfected HeLa cells increases in proportion to protein concentration in the range of 2 to 20 $\mu\text{g}/\text{mL}$, and the internalization is saturated at higher protein concentrations. The results were added to Supplementary Fig. 15 in the revised supporting information.

According to your suggestion, we tested the efficacies of TAT-conjugated BSA-FITC (each BSA was conjugated with 2, 4 or 6 TAT ligands using a SMCC linker) at a protein concentration of 10 and 20 $\mu\text{g}/\text{mL}$, respectively. As shown in Figure R5, the cells treated with TAT-conjugated BSA-FITC showed weak fluorescence. The protein delivery efficacy is much lower than those of fluoroamphiphiles shown in Figure R4. Besides TAT-conjugated BSA, the peptide GRKKRRQRREKIKRPRSSNAETL in Fig. 5c is also a TAT-conjugate. GRKKRRQRRR is the TAT sequence and EKIKRPRSSNAETL is a membrane-impermeable peptide. The designed peptide showed poor cytosolic delivery at a concentration of 5 $\mu\text{g}/\text{mL}$ (1 μg in each well), however, it exhibited efficient internalization in the presence of F4-1 or F3-3, suggesting that the fluoroamphiphiles have high activity in cytosolic delivery. The comparison of fluoroamphiphiles with TAT-conjugates was added to Supplementary Fig. 13 in the revised supporting information. Thanks very much for your suggestion.

Figure R4. (a) Fluorescence of HeLa cells treated with F4-1/BSA-FITC or F3-3/BSA-FITC complexes for 4 h at different protein concentrations analyzed by flow cytometry. (b) Confocal images of HeLa cells treated with F4-1/BSA-FITC or F3-3/BSA-FITC complexes for 4 h. The polymer concentrations for F4-1 and F3-3 are 10 and 12.5 $\mu\text{g/mL}$, respectively.

Figure R5. Fluorescence images of HeLa cells treated with TAT-conjugated BSA-FITC for 4 h. TAT peptide with a N-terminal cysteine was conjugated to BSA-FITC using a succinimidyl 4-(N-maleimidomethyl) cyclohexane-1-carboxylate (SMCC) linker. BSA-FITC was conjugated with 2, 4, or 6 TAT chains, respectively. The BSA concentration is 10 or 20 $\mu\text{g/mL}$.

3. I would appreciate if the applicability to other cell lines was presented.

Replies: According to your nice suggestion, we have tested efficacies of the lead materials F4-1 and F3-3 in protein delivery on several cell lines. As shown in Figure

R6, the materials showed high efficacies in the delivery of BSA-FITC into NIH3T3 and HEK293 cells. These results suggested that the fluoroamphiphiles are applicable to other cell lines. These results and related discussions were added to Fig. 5a, 5b in the revised main manuscript. Thanks very much for your nice suggestion.

Figure R6. Confocal images of NIH3T3 (a) and HEK293 (b) cells treated with F4-1/BSA-FITC and F3-3/BSA-FITC complexes for 4 h, respectively.

4. Even though the authors apply their system to two proteins and two peptides, complex formation is investigated only for BSA. Information on nanoparticle formation and the robustness of nanoparticle formation to different ratios should be presented. How much effort is needed to adapt the delivery approach to a particular experimental system?

Replies: According to your comment, we tested the nanoparticle formation (size and PDI) for BSA, β -gal, and saporin at different weight ratios. As shown in Table R1, both fluoroamphiphiles form uniform nanoparticles (around 200 nm, PDI < 0.3) with proteins within a wide range of protein to polymer weight ratios. The results suggest the robustness of nanoparticle formation at different ratios. The results were added as Supplementary Table 2 in the revised manuscript. We thank the reviewer for this useful suggestion.

BSA : F4-1	2	1.5	1	0.5	0.2	0.1
Size(nm)	284±10.7	213±2.7	190±4.2	227±7.1	238±4.1	265±5.1
PDI	0.02	0.16	0.19	0.25	0.12	0.26
BSA : F3-3	1	0.75	0.5	0.2	0.15	0.1
Size(nm)	166±3.9	189±7.2	209±5.4	235±9	249±11	282±12
PDI	0.17	0.19	0.18	0.22	0.23	0.3
β-gal : F4-1	2	1.5	1	0.5	0.2	0.1
Size(nm)	251.4±7.1	234±8	275±48	252±16	221±6.5	222±3.2
PDI	0.13	0.11	0.23	0.18	0.2	0.14
β-gal : F3-3	1.5	1	0.5	0.2	0.15	0.1
Size(nm)	239±7.1	265±3.9	230±17	236±14	238±8	256±13
PDI	0.09	0.2	0.22	0.23	0.23	0.3
Saporin : F4-1	3	2.5	2	1.5	1	0.5
Size(nm)	262±8	175±0.9	145±4.3	103±2.4	112±3	152±7
PDI	0.11	0.2	0.18	0.3	0.23	0.27
Saporin : F3-3	3	2.5	2	1.5	1	0.5
Size(nm)	272±8.4	172±2.3	187±3.5	165±7.4	172±1.7	225±0.2
PDI	0.05	0.1	0.09	0.16	0.26	0.3

Table R1. Size and PDI values of F4-1/protein and F3-3/protein complexes at different protein to polymer weight ratios.

5. The cellular delivery of beta-Gal followed by X-gal staining is not a proof for the full preservation of activity. The staining could as well result from a minor fraction of non-denatured protein. Would it be feasible to titrate beta-Gal with the amphiphile and determine the maintenance of activity in an *in vitro* system?

Replies: Thanks for the comments! Besides X-gal staining to confirm the β-Gal activity, we also quantitatively tested the β-Gal activity in the transfected cells by a β-galactosidase assay kit. O-nitrophenyl-β-D-galactopyranoside was used as the enzyme substrate. As shown in Figure R7, more than 95% of the activities of β-Gal delivered by F4-1 and F3-3 were recovered in the transfected cells after 4 h (Fig. 5f in the main manuscript), suggesting that most of the protein activity is maintained after cytosolic delivery. CD spectra in Figure R8 also showed that β-Gal in the formed complexes was mainly in native state rather than denatured form.

According to your suggestion, we also titrated β-Gal with F4-1 and F3-3 and then determined the enzyme activity in an *in vitro* system. As shown in Figure R9, the addition of fluoroamphiphiles into the β-Gal solution decreased the enzymatic activity to hydrolyze X-gal, however, the release of β-Gal from the complex by a polyanionic heparin sodium can fully recover the enzyme activity. The results suggest that the protein activity can be maintained when delivering by fluoroamphiphiles. The results on the maintenance of β-Gal activity were added as Supplementary Fig. 14 in the revised manuscript.

Figure R7. Relative β-Gal activity in HeLa cells transfected with F4-1/β-Gal, F3-3/β-Gal, and β-Gal alone for 4 h. 5 μg β-Gal and 2 μg F4-1 or 2.5 μg F3-3 were used in each well.

Figure R8. Circular dichroism spectra of F4-1/β-Gal and F3-3/β-Gal complexes, and β-Gal alone in aqueous solution.

Figure R9. Relative β -Gal activity in the presence of F4-1 and F3-3 (40 $\mu\text{g/mL}$), respectively determined by X-gal staining. The recovery of β -Gal activity was conducted by the addition of 0.1 mg/mL heparin sodium into the complex solutions. Data are presented as mean \pm s.e.m. ($n = 3$). ^{N.S.} $p > 0.05$ and ^{***} $p < 0.001$ analyzed by Student's t-test, one tailed.

6. For the proteins, also a concentration series for delivery is required. It could be that at low concentrations, the complexes disassemble before cellular entry. On the other hand, is it guaranteed that the complexes dissociate once inside the cell?

Replies: According to your suggestion, we tested the activities of lead materials F4-1 and F3-3 at different BSA-FITC concentrations. The protein concentration ranges from 2 to 40 $\mu\text{g/mL}$. As shown in Figure R10, the fluorescence from transfected HeLa cells increases in proportion to protein concentration in the range of 2-20 $\mu\text{g/mL}$, and the internalization is saturated at higher protein concentrations. The fluoroamphiphiles form uniform nanoparticles with the proteins within a relatively wide range of protein to polymer weight ratios (Table R1). Further increase in protein concentration does not reduce the fluorescence intensity in the transfected cells. The release of proteins from the complexes is guaranteed at different protein concentrations. As shown in Figure R3 and Figure R10, F4-1 and F3-3 are capable of rapidly transporting proteins into the treated cells and releasing the cargos into cytosol after endocytosis. The results were added as Supplementary Fig. 15 in the revised supporting information. Thanks very much for your valuable suggestion on this issue.

Figure R10. (a) Fluorescence of HeLa cells treated with F4-1/BSA-FITC or F3-3/BSA-FITC complexes for 4 h at different protein concentrations analyzed by flow cytometry. (b) Confocal images of HeLa cells treated with F4-1/BSA-FITC or F3-3/BSA-FITC complexes for 4 h. The polymer concentrations for F4-1 and F3-3 are 10 and 12.5 $\mu\text{g/mL}$, respectively.

7. The element mapping analysis would provide the means to understand to which degree complexation is a function of the molar ratio of the two components.

Replies: Thanks for the nice suggestion! Element mapping analysis cannot reveal the degree of protein complexation in the formed nanocomplexes because part of the proteins were adsorbed on the nanocomplex surface via ionic interactions, and this degree cannot be analyzed according to the mapping result.

However, we measured the complexation degree of BSA in the complexes of F4-1/BSA and F3-3/BSA at different polymer/protein molar ratios by a BCA assay. The proteins complexed by the fluoroamphiphiles cannot be measured by the assay, and thus we measured the protein contents in the complex solutions at different molar ratios of the two components. As shown in Figure R11, the complexation degrees of BSA in the complexes increased with increasing polymer/BSA molar ratio. This is reasonable due to saturated protein binding by the fluoroamphiphiles at higher feeding ratios.

Figure R11. Complexation degrees of BSA in the F4-1/BSA and F3-3/BSA complexes at different fluoroamphiphile/BSA molar ratios.

Though the element mapping results cannot provide information on complexation degree, we can estimate the molar ratio of protein to polymer in the complex. Take F4-1/BSA-Pt for example, the molar ratio of Pt to F in the nanoparticle is around 0.43:1 in the element mapping. The Pt nanoparticle has an average size of 2.05 nm (measured from 200 Pt nanoparticles), and each Pt particle has an approximate number of 130 Pt atoms. F4-1 has 390 fluorine atoms according to the fluorine element analysis. Thus, the molar ratio of BSA-Pt and F4-1 within the square frame of Figure S12 is calculated to be 1.29:1. Considering that 66% of BSA was complexed with F4-1 determined by the BCA assay (Fig. 4d), the molar ratio of complexed BSA-Pt to F4-1 is estimated to be 0.85:1. Thanks very much for your valuable suggestion.

Minor points:

8. Abstract, l. 11: instead of Fluorous ligands it should read fluororous substituents.

Replies: According to your suggestion, we have replaced the fluororous ligands by fluororous substituents.

9. Introduction: l. 37: I do not know of evidence that CPP-mediated delivery requires the denaturation of the cargo protein. Moreover, CPPs have been presented that work by non-covalent complexation (Biochim Biophys Acta. 2010 Dec; 1798(12):2274-85). Thus covalent modification is not required in all cases.

Replies: Thanks for the comments! We agree with you that covalent modification is not required in all the CPP-mediated protein delivery systems. Therefore, we changed our description on this issue in the revised manuscript to “most of them require chemical modification of target proteins or peptides”. For CPP “requires the denaturation of the cargo protein”, this is not a problem for all the CPP-mediated

delivery systems. The sentence was changed to “The covalent conjugation of PTD to proteins might be involved with sophisticated syntheses, reduced bioactivity and safety concerns”.

Thank you very much for your great comments and helpful suggestions to improve our manuscript! We appreciate your time and efforts paid on reviewing our manuscript.

Reviewer #2 (Remarks to the Author):

Review on the manuscript entitled: “The fluorous effect of fluoroamphiphiles in cytosolic protein delivery” by Zhang et al.

This paper reports on the potential of fluorinated amphiphiles in the cytosolic delivery of proteins. The fluorinated amphiphiles are co-assembled to form nanoparticles in aqueous solutions. This work is based on the fact that the combined hydrophobic and lipophobic characters of fluorinated chains improve the affinity of polymers for cell membranes and their penetration across lipid bilayers, thus facilitating the endocytosis for protein delivery. A series of fluoroamphiphiles was obtained by grafting various fluoroalkyl chains to a cationic poly(ethylamine) (PEI) polymer, and their capacity to deliver proteins and peptides (bovine serum albumin, β -galactosidase, saporin and the peptide GRKKRRQRREKIKRPRSSNAETL) was investigated. Their ability to deliver these molecules was compared to that of traditional amphiphiles such as alkane and cycloalkane-grafted PEI. The results show that the fluoroamphiphiles have definite advantages over the non-fluorinated amphiphiles, such as improved protein encapsulation, lower protein denaturation, facilitated cellular uptake and limited toxicity. A structure-activity relationship investigation also revealed that a balance in the fluorophilic character of the fluoroamphiphiles is beneficial for efficient protein delivery.

The paper is clearly organized and written, and adequately referenced. The methods are appropriate, and the results carefully interpreted.

The only objection that could be raised is that fluorinated amphiphiles degrade eventually into carboxylic acids that are persistent to the environment and toxic (see Krafft and Riess, *Curr. Opin. Colloid Interface Sci.* 2015, 20, 192). The concentrations used in this particular application are minute however, but the authors may consider adding a comment in order to lift a potential objection.

To my opinion this work provides a clear breakthrough in the field and should interest the broad readership of *Nature Communications*. I recommend publication with minor modifications.

Replies: Thanks very much for your nice comments on our manuscript. According to your suggestion, we added the following discussion on environmental challenges of fluorinated materials in the discussion section. “Though polyfluorinated substances were listed as persistent and bioaccumulative materials (Krafft and Riess, *Curr. Opin. Colloid Interface Sci.*, 2015, 20, 192-212), the concentrations of fluoroamphiphiles used for cytosolic protein delivery are minute, and further design of biodegradable

fluoroamphiphiles may resolve the bioaccumulation issues when the materials are used for *in vivo* protein delivery.

Typos:

Line 57: “fluoroamphiphiles”

Line 72: “cycloalkanes”

Figure 1 caption: “cycloalkane”

Replies: We have corrected these typing errors in the revised manuscript. In addition, we checked throughout the manuscript to avoid typos and grammar errors. Thank you very much for the comments on our manuscript! We greatly thank you for your time and efforts paid on reviewing our manuscript.

Reviewer #3 (Remarks to the Author):

In the manuscript “The fluorous effect of fluoroamphiphiles in cytosolic protein delivery”, Cheng and coworkers report the preparation of fluoroamphiphile/protein nanocomplexes for effective protein delivery into cells. Their characterization of these assemblies and the structure-activity relationship studies are thorough and make for a complete paper. The data presented in Figure 2 clearly show their strategy is successful. The use of fluorous amphiphiles for protein delivery is novel, and is appropriate for publication in Nature Communications once the following comments are addressed:

Replies: Thanks very much for your nice comments and suggestions on our manuscript! We have replied to your detailed comments one-by-one as follows.

1. Library Synthesis and Initial Comparisons

(1) It should be explicitly stated in the text that branched PEI was used to make the various polymer amphiphiles. Figure 1a should also be modified to show the branched structure of the polymer being modified with the fluorous ligands. It should also more realistically schematize the nanocomplex.

Replies: Thanks for the comment! We now stated that branched PEI was used to synthesize the polymer amphiphiles in the main manuscript. Also, we changed the linear structure of PEI to a branched one in Figure 1a to avoid confusion. Thanks for the valuable suggestion.

(2) Number of conjugated fluorous ligands is given as a number average. It would be useful for weight% fluorine to be provided in Figure 5 or Table S1, as this is standard for characterizing fluorous compounds and makes for easier comparison between the different materials in the F series.

Replies: According to your suggestion, we have added the fluorine content (wt%) of each fluoroamphiphile in the revised Table S1.

(3) If known, it would be useful to describe/show the general structure of commercial PULSinTM and how it compares to their polymer material.

Replies: PULSinTM is a cationic lipid-based formulation for protein and peptide delivery developed by Polyplus-transfection. The general structure and component of PULSinTM is a commercial secret. PULSinTM-mediated BSA-FITC delivery was conducted according to the manufacture's protocol. The following information was stated in the methods section. "Commercial protein transfection reagent PULSinTM was used as a positive control, and the transfection was conducted according to the manufacture's protocol. 4 μ L reagent was used for each well".

(4) The use of trypan blue as a quenching agent for the extracellular fluorescence should be referenced or, if a novel procedure, experimentally validated.

Replies: According to your request, we cited a reference on using trypan blue as a quenching agent for extracellular fluorescence in the revised manuscript (Cytometry Part A, 2005, 65A, 93-102). Thanks for the nice suggestion!

2. Cell Culture and Biomolecule Delivery

(1) Delivery was nicely optimized with BSA-FITC, and then translated to other enzymes to show activity after delivery. However, since cytosolic protein delivery is being claimed in the paper and the title, the authors should provide insight into how their material escapes the endosome and how the protein is released from the polymer. If the experiments in Figure 2 are meant to depict endosomal escape, that should be explicitly stated in the text. Endosomal escape could also be shown via co-localization with an endosomal stain.

Replies: According to your suggestion, we investigated the time-dependent cellular uptake and intracellular trafficking of fluoroamphiphile/BSA-FITC complexes and non-fluorinated controls. The acidic organelles in the cells were stained with LysoTracker Red. As shown in Figure R12, HeLa cells treated with F4-1/BSA-FITC and F3-3/BSA-FITC showed strong and evenly distributed fluorescence in the cytosol after 1 h incubation, and the internalized proteins (FITC, green) were not co-localized with acidic organelles (LysoTracker Red, red). In comparison, cells incubated with the control complexes exhibited weak and un-internalized green dots on the cell surface. We further showed that F4-1 and F3-3 complexes are capable of releasing BSA-FITC in the presence of heparin sodium and unlabeled BSA (mimic polyanions and proteins in cytosol). The release of BSA-FITC from the complexes is confirmed by competitive binding experiments (Figure R13). The complexation of BSA-FITC with the fluoroamphiphiles showed much decreased fluorescence due to aggregation of BSA-FITC in the complexes, while the addition of heparin sodium or BSA can fully or partially recover the quenched fluorescence from BSA-FITC. The results together suggested that F4-1 and F3-3 are capable of efficiently transporting proteins into the treated cells and releasing the cargos into cytosol after endocytosis. We appreciate the valuable suggestions. The supplemented results on co-localization of transported BSA-FITC with acidic organelles were added to the revised manuscript (Supplementary Figs. 4 and 6).

Figure R12. Confocal images of HeLa cells treated with different polymer/BSA-FITC complexes for 0.5 h, 1 h, 2 h, 4 h, 6 h and 8 h, respectively. The acidic organelles in HeLa cells were stained with LysoTracker Red.

Figure R13. Fluorescence intensity of F4-1/BSA-FITC (a) and F3-3/BSA-FITC (b) complexes in the absence and presence of heparin sodium (0.5 mg/mL) or native BSA (5 mg/mL). BSA-FITC alone was tested as a control.

(2) Insight into how the fluoroamphiphile cationic polymers efficiently encapsulate positively charged molecules (saporin and the peptide) should be included.

Replies: The fluoroamphiphiles F4-1 and F3-3 failed to assemble into nanoparticles in the absence of proteins at concentrations up to 15 μ M, however, both F4-1 and F3-3 form uniform nanoparticles (around 200 nm, PDI < 0.3) with saporin within a range of saporin to polymer weight ratios (Table R2), suggesting that saporin facilitated the assembly of fluoroamphiphiles in aqueous solution. Though saporin is a positively charged protein, the binding of anionic region of the protein to cationic PEI of fluoroamphiphiles may help the assembly process. We added these discussions to the revised manuscript. Thanks very much for your nice suggestion.

Saporin : F4-1	3	2.5	2	1.5	1	0.5
Size(nm)	262±8	175±0.9	145±4.3	103±2.4	112±3	152±7
PDI	0.11	0.2	0.18	0.3	0.23	0.27

Saporin : F3-3	3	2.5	2	1.5	1	0.5
Size(nm)	272±8.4	172±2.3	187±3.5	165±7.4	172±1.7	225±0.2
PDI	0.05	0.1	0.09	0.16	0.26	0.3

Table R2. Size and PDI values of F4-1/saporin and F3-3/saporin complexes at different saporin to polymer weight ratios.

(3) The peptide sequence should be listed in the Figure 2 legend.

Replies: We have added the peptide sequence in the figure legend (now Fig. 5) according to your suggestion.

3. Comparison of Fluoroamphiphiles and Non-Fluorinated Controls in Protein Delivery

In line 116, the authors write that “the fluoroamphiphiles and phospholipids in cell membranes have limited miscibility, which ensures efficient cell internalization of fluoroamphiphile-based nanocomplexes.” This should have a citation (K. Kumar work is relevant)

Replies: We added two citations of K. Kumar’ work (Kumar K., J. Am. Chem. Soc., 2007, 129, 9037-9043; J. Am. Chem. Soc., 2009, 131, 12091-12093) to prove the limited miscibility of fluorinated and hydrocarbon lipids according to your suggestion.

4. Nanocomplex Formation, Protein Denaturation, Cellular Uptake, and Cytotoxicity of Fluoroamphiphiles and Non-Fluorinated Controls and Structure-Activity Relationships

(1) The zeta-potentials of A3-3 and A4-1 nanocomplexes should be added to Figure 4a.

Replies: We have added the zeta-potentials of F4-1, F3-3, A3-3 and A4-1 nanocomplexes to the revised Figure (now Fig. 3a) according to your suggestion.

(2) The difference between materials F2 and F3 (a single hydrogen) should be commented on as the effect on protein delivery is striking.

Replies: According to your suggestion, we added the following discussion in the revised manuscript to comment on this issue.

“In the library, the fluoroalkanes F1, F2, F3 and F4 have 7, 8, 9 and 13 fluorine atoms in the structure, respectively. The striking difference between F1/F2- and F3-based fluoroamphiphiles in the delivery of BSA-FITC can be explained by an additive fluorous effect. A single or two fluorine differences in the structure may have

significant influence on its physicochemical properties and transfection efficacy. Therefore, the length of fluoroalkyl chain and the fluorination degree on a specific fluoroamphiphile need to be optimized before use.”

(3) This discussion would greatly be enhanced with a schematic showing the proposed assembly structures. (F4-1 and F3-3 compared to F4-4 and F3-4 in Figure 5c).

Replies: According to your suggestion, we added schemes for the proposed assembly structures to the revised figure (Figure R14). The proposed models for F4-1/F3-3 and F4-4/F3-4 complexes with BSA are shown in top left and bottom left panels of (c), respectively.

Figure R14. Structure-activity relationships of fluoroamphiphiles in cytosolic protein delivery. The proposed models for F4-1/F3-3 and F4-4/F3-4 complexes with BSA are shown in top left and bottom left panels of (c), respectively.

5. Other comments

(1) The term “fluorous effect” which is used in the title and throughout the paper is an existing term in the literature with a specific definition referring to the fact that highly fluorinated molecules “exhibit an unusual propensity to phase segregate [e]” (Marsh, E.N.G. *Acc. Chem. Res.* 2014, 47, 2878–2886; Marsh, E.N.G. and coworkers, *Biochemistry* 2004, 43, 16277-16284) or to the “observation that highly fluorinated or perfluorinated compounds have a tendency to exclude themselves from both aqueous and organic phases” (Clark, A.W. and coworkers, *Chem. Commun.* 2017, 53, 3094; Horváth, I.T.; Curran, D.P.; Gladysz, J.A. *Handbook of Fluorous Chemistry*, Wiley-VCH Verlag GmbH & Co. KGaA, 2005, pp. 1–4).

Replies: Thanks very much for pointing out the definition of fluorous effect in previous references. We agree with you that the fluorous effect is an existing term. To

avoid confusion, we avoided to re-define the fluorous effect in the revised manuscript. The sentence “the fluorous effect was defined as unique performance of fluoroamphiphiles in protein delivery in comparison with non-fluorinated control materials” was removed.

(2) The authors should refrain from redefining “the fluorous effect” as they do in line 101, and perhaps instead use the wording “the effect of fluorination...”

Replies: we have changed phrase to “the effect of fluorination” in line 101 according to your suggestion.

(3) The work in this paper would be better presented by beginning with the structures of the nanocomplexes and how those affect protein denaturation, membrane permeability, and cytotoxicity. Figure 4 should come before Figure 3 (characterization before application).

Replies: We agree with you that characterization usually comes before application for biomaterials. The case in this manuscript is a little different. We screened high efficient polymers for protein delivery in a library of fluoroamphiphiles and non-fluorinated controls. Distinct behaviors between fluoroamphiphiles and non-fluorinated controls were observed. Then, characterizations on nanoparticle formation, protein denaturation, membrane permeability, and toxicity were conducted to explain the differences and conclude the effect of fluorination. This writing logic is easier for the readers. Thanks very much for your suggestion on this issue.

(4) Figure 2 is the most impressive figure, which shows how effective their scaffold is; as such, it should be the final figure of the paper.

Replies: According to your suggestion, we changed Fig. 2 to the final figure of this manuscript (now Fig. 5). We agree with you that it is better to show how effective the scaffold is at the end of manuscript.

(5) There are numerous typos and grammar errors, which should be corrected.

Replies: Thanks very much for your suggestion! We have carefully checked the typos and grammar errors throughout the manuscript. Thank you very much for the valuable comments to improve our manuscript! We greatly thank you for your time and efforts paid on reviewing our manuscript.

Reviewers' comments:

Reviewer #1 (Remarks to the Author):

I compliment the authors with their efforts to experimentally address the reviewers' concerns. A considerable number of extra experiments have been conducted which strongly improve the quality of the manuscript. Some minor comments remain which primarily relate to the quality of microscopy images.

I appreciate the efforts in conducting additional experiments with endocytosis inhibitors and time-lapse analysis of protein localization. Taken together the data suggest endocytosis with rapid endosomal release. I would appreciate if enlargements of the microscopy images are provided. At this point, it is impossible to recognize the structures of the lysosomes.

I have my doubts on the microscopy data presented for the TAT conjugates. The images are of very low quality and look as out of focus. To eliminate these doubts, the authors may want to also present transmission images. For a valid comparison, the TAT conjugates should be presented along with the polymers in one experiment.

Protein concentrations should also be provided in molar concentrations.

Reviewer #2 (Remarks to the Author):

I have thoroughly read the corrected version, and found the manuscript significantly improved. The authors have satisfactorily answered to the questions that were raised. I now consider that this work deserves publication in Nat. Comm.

Reviewer #3 (Remarks to the Author):

Zhang et. al have largely addressed the concerns and this work is appropriate for publication in Nature Communications.

The only remaining concern is the use of the "fluorous effect" is still incorrect within the manuscript. The authors do not disagree with the comment that the fluorous effect is an existing term and they removed their redefinition statement yet continue to use "fluorous effect" in the manner of their redefinition.

Additionally, this manuscript still needs editing before publication as there are a number of instances where the word choice is incorrect.

Responses to the Referees

Reviewer #1 (Remarks to the Author):

I compliment the authors with their efforts to experimentally address the reviewers' concerns. A considerable number of extra experiments have been conducted which strongly improve the quality of the manuscript. Some minor comments remain which primarily relate to the quality of microscopy images.

Replies to the Comments: Thanks very much for your nice comments on our manuscript. The detailed comments are replied one-by-one as follows. We appreciate your time contributed on reviewing the manuscript.

I appreciate the efforts in conducting additional experiments with endocytosis inhibitors and time-lapse analysis of protein localization. Taken together the data suggest endocytosis with rapid endosomal release. I would appreciate if enlargements of the microscopy images are provided. At this point, it is impossible to recognize the structures of the lysosomes.

Replies to the Comments: Thanks for the comments! According to your suggestion, we enlarged the confocal images of F4-1/BSA-FITC and F3-3/BSA-FITC complexes to provide high-resolution confocal image (Figure R1). Now, it is possible to recognize the structure of endolysosomes in the transfected cells. The enlarged confocal images were updated as supplementary Figure 4 and supplementary Figure 6 in the revised supporting information. Thanks very much for your nice suggestions on this issue.

Figure R1. Confocal images of cells treated with fluoroamphiphile/BSA-FITC complexes. HeLa cells were incubated with the complexes for 0.5 h, 1 h, 2 h, 4 h, 6 h and 8 h, respectively. 4 μg BSA-FITC (0.3 μM) was complexed with 2 μg F4-1 and 2.5 μg F3-3, respectively. The acidic organelles in HeLa cells were stained with LysoTracker Red. The scale bar is 25 μm .

I have my doubts on the microscopy data presented for the TAT conjugates. The images are of very low quality and look as out of focus. To eliminate these doubts, the authors may want to also present transmission images. For a valid comparison, the TAT conjugates should be presented along with the polymers in one experiment.

Replies to the Comments: According to your suggestion, we now provided confocal images of cells treated with F4-1/BSA-FITC, F3-3/BSA-FITC and TAT-conjugated BSA-FITC for 4 h. The TAT conjugates were shown along with the polymers F4-1

and F3-3 in one experiment according to your suggestion. In addition, we quantitatively analyzed the cytosolic delivery of BSA-FITC by fluoroamphiphiles and TAT conjugates by flow cytometry. As shown in **Figure R2**, the polymers F4-1 and F3-3 are much more efficient than the TAT conjugates in the delivery of BSA-FITC. The suggested figures are updated as supplementary Figure S13 in the revised supporting information. Thanks very much for your nice suggestion.

Figure R2. Efficacies of F4-1/BSA-FITC, F3-3/BSA-FITC and TAT-conjugated BSA-FITC in cytosolic delivery. (a) Confocal images of HeLa cells treated with F4-1/BSA-FITC, F3-3/BSA-FITC and TAT-conjugated BSA-FITC for 4 h. TAT peptide with a N-terminal cysteine was conjugated to BSA-FITC using a SMCC linker. The BSA-FITC was conjugated with 2, 4, or 6 TAT chains, respectively. The BSA concentration is 20 µg/mL. The scale bar is 50 µm. (b, c) Fluorescence intensity of HeLa cells treated with F4-1/BSA-FITC, F3-3/BSA-FITC and TAT-conjugated BSA-FITC for 4 h at BSA concentrations of 10 µg/mL (0.15 µM, b) and 20 µg/mL (0.3 µM, c), respectively analyzed by flow cytometry.

Protein concentrations should also be provided in molar concentrations.

Replies to the Comments: We have added the molar concentrations of proteins in the revised manuscript according to your nice suggestion. Again, your time paid on reviewing our manuscript is highly appreciated.

Reviewer #2 (Remarks to the Author):

I have thoroughly read the corrected version, and found the manuscript significantly improved. The authors have satisfactorily answered to the questions that were raised. I now consider that this work deserves publication in Nat. Comm.

Replies to the Comments: Thanks very much for you nice comments on our manuscript.

Reviewer #3 (Remarks to the Author):

Zhang et al. have largely addressed the concerns and this work is appropriate for publication in Nature Communications.

Replies to the Comments: Thanks very much for your comments on our manuscript. The detailed comments are replied one-by-one as follows.

The only remaining concern is the use of the "fluorous effect" is still incorrect within the manuscript. The authors do not disagree with the comment that the fluorous effect is an existing term and they removed their redefinition statement yet continue to use "fluorous effect" in the manner of their redefinition.

Replies to the Comments: Thanks very much for your suggestion! We have changed the term of "fluorous effect" to "the effect of fluorination" or "fluorination effect" in the revised manuscript according to your suggestion.

Additionally, this manuscript still needs editing before publication as there are a number of instances where the word choice is incorrect.

Replies to the Comments: We have carefully checked the language and typing of the manuscript according to your suggestion. Again, we appreciate your time contributed on reviewing the manuscript.

REVIEWERS' COMMENTS:

Reviewer #1 (Remarks to the Author):

The authors have adequately responded to all comments. I recommend the manuscript for acceptance.

Reviewer #3 (Remarks to the Author):

The authors have addressed the last remaining concerns and this reviewer can now recommend publication.